# Few-shot Fine-grained Image Classification with Interpretable Prompt Learning through Distribution Alignment

## Abstract

Explainable few-shot fine-grained image classification is an essential task to align AI with human preferences by enabling precise recognition of subtle differences and providing explanations for decisions. Existing supervised models often struggle in few-shot scenarios due to their reliance on extensive labeled data, which is intractable to collect for customized human preferences. Meanwhile, large vision-language models (VLMs) while robust in zero-shot tasks, fail to capture the subtle difference required for fine-grained classification. In this work, we introduce a novel approach that enhances AI alignment in both zero-shot and few-shot fine-grained image classification by leveraging explainable prompt learning and distribution alignment techniques. Specifically, we utilize pre-trained LLM to expand the label space in a training-free manner, addressing the disparity between plain text and the image-text corpus distributions. This is further enhanced by a few-shot learning pipeline that incorporates prompt learning with a weighted distribution alignment mechanism between image and text representations for better alignment with human-like understanding. The proposed approach not only addresses the limitations of current prompting techniques but also enhances interpretability. Extensive experiments demonstrate the effectiveness of our method and illustrate the interpretability of our descriptions.

## 1 Introduction

Fine-grained image classification is a fundamental task for aligning AI with human preferences, particularly when the AI must focus on subtle differences between images and categories, such as distinguishing between different species of birds or varieties of flowers. This capability is crucial for applications where precision and sensitivity to detail are necessary, enabling AI to deliver outcomes that closely match human expectations. Moreover, improving the AI's ability to detect these subtle distinctions enhances its capacity to make decisions that are more personalized and contextually appropriate. Equally important is the ability to explain its classifications and decisions, providing transparency by helping users understand how and why specific choices are made. This combination of precise classification and clear explanation is essential for ensuring that AI systems align effectively with the user preferences.

Large Vision-Language Models (VLMs), such as CLIP (Radford et al., 2021), have demonstrated remarkable capabilities in zero-shot image classification tasks, showcasing their potential in aligning AI systems with complex human visual preferences. However, these models require finely engineered prompts to mitigate the gap between text and image distributions. Despite the advancements in prompt learning methods (Zhou et al., 2022b;a; Menon & Vondrick, 2023), which aim to refine prompts to better match specific image categories, significant challenges remain. Current discrete prompt learning approaches often fail to dynamically align the image and text distributions for specific datasets, resulting in sub-optimal performance. Moreover, soft prompting methods, such as COOP (Zhou et al., 2022b), lack the ability to provide explanations for their reasoning processes, which is crucial for transparency and trust in AI applications.

Our research identifies a notable distribution discrepancy Sun et al. (2023) as depicted in Figure 1, which is prevalent across both current prompt learning models and broader vision-language models. This misalignment, not only between the text corpus and the image-text corpus but also directly within the image-text

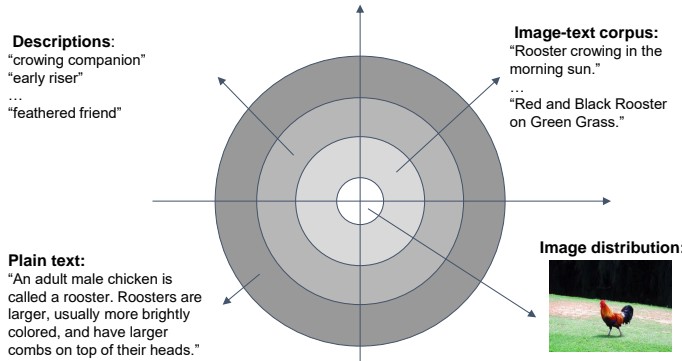

**Descriptions**:
"crowing companion"
"early riser"
…
"feathered friend"

**Image-text corpus:**
"Rooster crowing in the morning sun."
…
"Red and Black Rooster on Green Grass."

**Plain text:**
"An adult male chicken is called a rooster. Roosters are larger, usually more brightly colored, and have larger combs on top of their heads."

**Image distribution:**

Figure 1: An illustration of the distribution gap between images and different types of texts. The coordinate represents the embedding space. Each circle and ring represent their distributions. For example, the white circle indicates the image distribution and the first outer ring represents the image-text corpus distribution. The closer to the origin, the more similar distributions they have.

interactions, poses significant challenges for the performance and reliability of VLMs like CLIP. Addressing this gap is essential for improving model performance and ensuring that AI systems can effectively interpret and align with human needs.

In order to jointly tackle those challenges inherent in diverse prompting techniques, we propose a novel training-free strategy to minimize the shift in text distribution between plain text and image-text corpus. We first leverage the latent knowledge inherent in LLMs, such as GPT (Brown et al., 2020), to mitigate the distribution shift between the text corpus and the image-text corpus. These models have demonstrated remarkable capabilities in generating various texts, making them promising candidates for effectively mitigating the distribution gap between various of texts. Thus, we design a prompting method on GPT-4 and GPT-4o, enabling them to generate semantic descriptions related to designated labels, while maintaining similarity to the image-text corpus that aligns with the distribution of the image-text model.

Furthermore, we present a few-shot weighted strategy incorporating a soft prompt learning methodology to further mitigate the distribution discrepancy between text and images. This approach adjusts the prompt weights based on their relevance to the target domain, ensuring a better alignment of the distributions and improving classification performance.

In summary, our contributions can be summarized as follows: (1) We design a zero-shot training-free strategy to generate descriptions that are interpretable and beneficial for image-text retriever. (2) We propose a framework that learns soft prompts dynamically aligning image and text distribution while preserving interpretability. (3) The extensive experiments show the effectiveness and interpretability of our method, providing insights that can be universally applied to enhance future model designs and training paradigms.

## 2 Related Work

### 2.1 Large Vision-Language Models

Large pre-trained models recently show great potential in representation learning, which have greatly advanced many downstream applications in natural language understanding and computer vision. Following the seminal work Transformer (Vaswani et al., 2017), many generative AI models emerge rapidly. The GPT series models (Radford et al., 2018; 2019; Brown et al., 2020; OpenAI, 2023; Touvron et al., 2023) have shown their powerful ability in text mining and reasoning. There are also many methods (Menon & Vondrick, 2023; Wei et al., 2021) that try to leverage the reasoning ability to improve downstream tasks. However, large language models are always trained on pure text data from Internet, so it is hard to align image distributions. On the other hand, large vision-language models (Radford et al., 2021; Li et al., 2022; 2023; Rombach et al., 2022; Jia et al., 2021; Singh et al., 2022; Liu et al., 2023; Chen et al., 2024; Yang et al., 2024) are able to bridge the images and texts in the latent space. Particularly, CLIP (Radford et al., 2021) is trained from a large set of image-text pairs, and it successfully mitigates the distribution discrepancy between text and images with the contrastive loss. It also shows tremendous zero-shot ability on the image classification task. However, its classification ability is highly dependent on the enormous training data. CLIP (Radford et al., 2021) takes the advantages of selected 400 million image-text pairs (Schuhmann et al., 2021), while

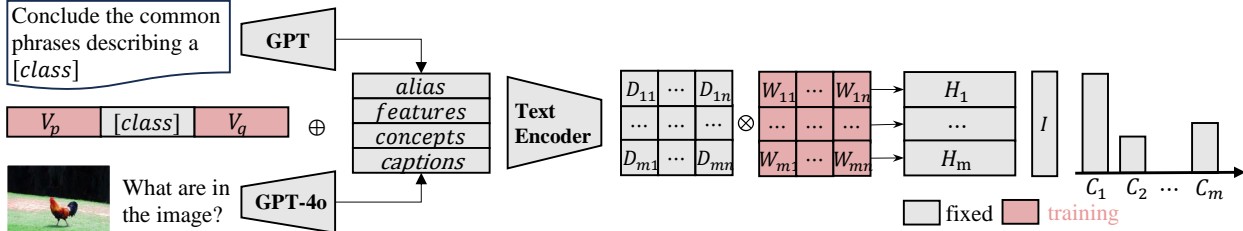

Figure 2: The overview of our proposed framework. Except for the learnable prompts $V_p, V_q$, and the learnable weights $W$, all other models, including GPT, GPT-4o, Text Encoder, are fixed. $D$ denotes the text embedding genereated by text encoder, and $H$ represents the weighted text embedding, and $C$ denotes the category. $m$ is the number of categories and $n$ is the number of descriptions for each category. $I$ represents the image embedding of a testing image. Initially, GPT and GPT-4o are prompted to generate descriptions for each category. We then learn soft prompts and learnable weights based on these descriptions to align the text distribution more closely with the image distribution, using the contrastive loss as a measure.

ALIGN (Jia et al., 2021) takes 1.8 billion noisy image-text pairs. This training paradigm is largely dependent on training data. In other words, if the training data cover the specific domain, such as animals, food, or daily appliances, the model can perform well. However, when it comes to some unseen or uncommon domains, such as handwriting digits or medical images, the model would have quite poor performance. It reveals that even in large pre-trained models, it still has the domain gap issue. To address that issue, we come up with a method which leverages the reasoning ability of large language models and vision language models to generate some descriptions that have semantic information, like features, alias, and some related concepts. In this way, it can mitigate the distribution shift between plain text category and image-text corpus, which is better aligned with the CLIP model distribution.

## 2.2 Prompt Learning

Prompt learning involves the creation of prompts that guide language models to better comprehend and respond to questions. The essence of prompt learning is to leverage the context-aware nature of language models, especially large pre-trained models like GPT, by providing a structured context that directs the model's generation capabilities towards a desired output. However when only a small amount of task-specific data is available, prompt learning emerges as an alternative where the model is given a prompt that includes instructions or an example of the desired task, effectively converting various NLP tasks into a text completion problem.

While prompt learning achieves great performance on model tuning and downstream tasks, it is still underexplored in computer vision and visual language modeling. Pre-trained visual language models are incorporated with text information, and according to recent work (Radford et al., 2021; Zhou et al., 2022b; Wang et al., 2023b; Udandarao et al., 2023; Ge et al., 2023; Chen et al., 2023; khattak et al., 2023), the text prompt also makes a difference in image classification. To exploit text prompts in the classification problem, discrete prompts (Zang et al., 2022; Radford et al., 2021; Menon & Vondrick, 2023) are adopted to infer domain-specific knowledge. However, those methods may meet the suboptimal issue, which may not align with the specific domain distribution. To address this problem, some soft prompt learning methods (Zhou et al., 2022b;a; Hantao Yao, 2023; Lu et al., 2022; Zhu et al., 2022; Gao et al., 2024; Zhang et al., 2022; Chen et al., 2022; Wang et al., 2023a; Yu et al., 2025; Shi et al., 2025; Xiao et al., 2025; Li et al., 2025; Zhu et al., 2025) have been proposed. They take a few image samples to guide the text prompt to align with the image distribution. For example, CLIP-adapter (Gao et al., 2024) adapts both image and text embeddings to a new embedding space. Xiao et al. (2025) propose a test-time prompt tuning method that enhances zero-shot generalization in vision-language models. DePT Zhang et al. (2024) decouples base specific knowledge from feature channels into an isolated feature space during prompt tuning. CoPrompt Roy & Etemad (2023) enforces the prediction consistency of the trainable and pre-trained models to prevent overfitting on the downstream task. COOP (Zhou et al., 2022b) is the most related work to ours, which expands the labels

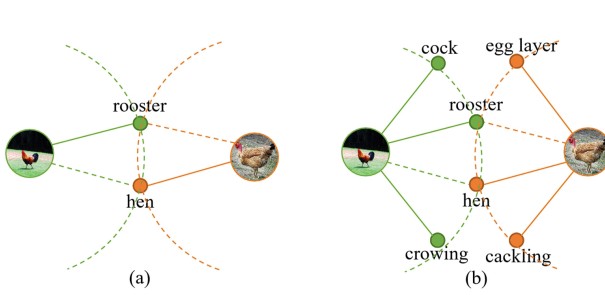

(a)          (b)

Figure 3: Illustration of how our descriptions work. (a) The original image-text retrieval solution is shown, where two images from different categories may have similar or even reversed similarity scores with their respective labels. For instance, an image of a rooster may have the same similarity score with both rooster and hen. (b) To address this issue, we expand the labels using their features, aliases, and concepts, leveraging prior knowledge and a closer image-text distribution. This approach leads to a higher error-tolerant rate, ultimately resulting in improved classification performance.

with several learnable prompts and optimizes them with few-shot image samples. However, discrete prompt learning methods may face the sub-optimal problem, while soft prompt learning methods have very limited interpretability. To overcome these issues, we propose a new method which jointly learns soft prompts while preserving the interpretability. We first generate some descriptions by a large language model that provides the prior knowledge and mitigates the pure text distribution and image-text distribution. Furthermore, we learn soft prompts to mitigate the distribution discrepancy between image distributions and text distributions. In this case, our method can preserve the prior knowledge of descriptions and provide explanations.

## 3 Approach

In this section, we first briefly introduce our motivation and the overview of our framework, which is illustrated in Figure 2. Then we present the technical details of our model in three main parts, including generating descriptions, prompt tuning, and weighted aggregation.

### 3.1 Overview

Our objective is to enhance image classification using existing large pre-trained model while simultaneously delivering interpretable explanations. To accomplish this, we introduce our framework, illustrated in Figure 2. This framework consists of two principal components. The first part is static description generation, which leverages the reasoning ability of large language models, such as GPT-4 (OpenAI, 2023) and GPT-4o to provide semantic description while aligning text distributions. The other part is the prompt learning pipeline. With the prior knowledge provided by the large language model, few-shot images further guide the learnable prompts and learnable weights to adapt the description distribution to align with the image distribution.

### 3.2 Generating Descriptions

Although the CLIP model shows its strong zero-shot ability, it still has domain bias due to the training data (Radford et al., 2021). Specifically, the distribution shift not only happens between images and texts, but also exists in natural languages. For example, a lot of natural language fine-grained categories, such as textures, landscapes, are less likely to appear in image-text corpus. The fact that the image-text corpus often comprises a higher frequency of keywords and a diminished occurrence of formal sentence structures reflects the distribution gap. (Schuhmann et al., 2021) In order to address the text distribution shift issue, we broaden the scope of labels by encompassing multiple descriptions which are more similar to image-text corpus and impart prior knowledge by leveraging the reasoning and generative ability of large language models. Figure 3 illustrates the expanding details.

Given a particular category, our approach involves harnessing the capabilities of large language models to generate a set of text descriptions, denoted as $D$. These descriptions enrich the label information and match the distribution found in the image-text corpus. This process can be formulated as: $D_c = g(c)$, where $g()$

represents any generic large language models, and $D_c$ refers to the descriptions regarding the category $c$. Descriptions will be used to enhance the effectiveness of the visual language classification model.

Basically, we use LLMs to conclude several common phrases that typically appear in image-text corpus. Inspired by the prompt work (Menon & Vondrick, 2023), we design a text prompt to guide GPT-4 to generate related descriptions as:

*Q: Conclude the common phrases in image-text pairs corpus that describe a {category}.*
*A: Some of the most common phrases used to describe a {category} include:*

For the possible few-shot images, we leverage the GPT-4o model to generate image features directly.

Furthermore, it is essential to anchor the generated descriptions to their corresponding labels. This is especially crucial in fine-grained classification tasks, where a single description might be applicable to multiple closely related categories. For example, "man's best friend" or " four legs" can describe all fine-grained animal categories, like different dogs and cats. Thus, to narrow its meaning, we formulate a training-free zero-shot prompt as *"an image of [class], which relates to [description]."*

### 3.3 Prompt Tuning

In our Prompt Tuning module, we aim to generate soft prompts for a set of images $X$ that can effectively bridge the domain gap between text and image distributions. Meanwhile, it should align with the latent distribution of CLIP. Consequently, as illustrated in Figure 2, we propose a description-based prompt tuning framework to reduce the distribution gap between the generated descriptions and images in a specific domain. Specifically, we introduce two soft prompts $V_m, V_n$ for the category label and its descriptions, respectively. Inspired by (Zhou et al., 2022b;a; Hantao Yao, 2023), we use the unified prompt for all classes and descriptions, which shares the soft prompts across all descriptions. The prompt is designed as:

$$t = [V_p][class][V_q][description], \tag{1}$$

where $V_p$ is the class soft prompt and $V_q$ is the description prompt. Each $v \in \{V_p, V_q\}$ has the same dimension as the word embedding. $V_p$ and $V_q$ have $p$ and $q$ soft prompts, respectively, where $p$ and $q$ are hyperparameters. In particular, each $t$ corresponds only to a single description.

Formally, we define the texture embedding encoded by the CLIP text encoder $\theta$ as $D_{ci} = \theta(t_{ci})$, where $c \in [0, m]$ refers to the class and $i \in [0, n]$ refers to the description $i$ corresponding to c. Additionally, $m$ is the number of categories and $n$ is the number of descriptions of one specific category. In this way, each prompted description will have one unique text embedding.

### 3.4 Weighted Aggregation

Existing works (Zhou et al., 2022b;a; Hantao Yao, 2023) primarily focus on prompting labels within a single text embedding. Nevertheless, our approach differs because we deal with scenarios where a single class could be described in multiple ways. This necessitates the aggregation of these description embeddings into a single class embedding to compute similarity scores with image embeddings.

However, descriptions are typically generic and might not align perfectly with all images across different domains or datasets. Therefore, a simple averaging of all descriptions could still introduce significant domain shift. In this case, we further mitigate the distribution difference between text descriptions and image distributions, specifically focusing on the few-shot target domain images denoted as $X$.

We propose enhancing category representation by emphasizing the precise text descriptions that have a high correlation with the image-text corpus. This allows us to compute a more accurate and interpretable similarity score for a category via weighted summation.

In other words, we introduce a learnable weight matrix $W = R^{m*n}$ as shown in Figure 2. The weight matrix has the same dimension as the description embedding matrix. To aggregate the description embeddings into

| model | Average | | | DTD | | | Food101 | | | OxfordPets | | |
|---|---|---|---|---|---|---|---|---|---|---|---|---|
| | CLIP | CBD | Ours | CLIP | CBD | Ours | CLIP | CBD | Ours | CLIP | CBD | Ours |
| RN50 | 51.2 | 51.8 | **52.8** | 40.3 | 38.6 | **42.0** | 74.5 | 74.9 | **76.8** | 82.2 | 81.8 | **83.2** |
| VITB32 | 55.3 | 55.7 | **56.8** | 43.8 | 43.1 | **45.5** | 78.3 | 78.9 | **80.3** | 83.0 | 84.9 | **85.5** |
| VITB16 | 59.5 | 60.4 | **61.2** | 44.4 | 46.3 | **46.9** | 85.2 | 84.3 | **85.9** | 86.9 | 86.6 | **87.3** |

| model | Imagenet | | | Flower102 | | | FVGCAircraft | | | EuroSAT | | |
|---|---|---|---|---|---|---|---|---|---|---|---|---|
| | CLIP | CBD | Ours | CLIP | CBD | Ours | CLIP | CBD | Ours | CLIP | CBD | Ours |
| RN50 | 58.2 | 58.9 | **59.5** | 60.5 | 63.1 | **63.8** | 17.00 | 17.3 | **17.6** | 25.5 | **28.2** | 26.5 |
| VITB32 | 61.8 | 62.8 | **63.1** | 63.5 | 64.1 | **66.1** | 18.6 | **19.4** | 19.3 | **38.4** | 36.6 | 38.0 |
| VITB16 | 67.4 | 67.7 | **68.3** | 68.6 | 69.7 | **71.5** | 23.0 | 23.1 | **24.3** | 40.9 | **45.4** | 44.2 |

Table 1: Comparison of our training free model with two baselines. We test our model on 7 datasets with 3 CLIP backbones. Our results show consistent improvement across all settings. The number indicates the image classification accuracy. The best results are shown in **bold**.

category embeddings, we perform a weighted summation as follows:

$$H_c = \sum_{i}^{n} D_{ci} \times \frac{W_{ci}}{\sum_{k=1}^{n} W_{ck}}, \qquad (2)$$

where $D$ is the description embeddings and $W$ is the weight matrix. Once we obtain the category embeddings $H$, we are able to calculate the similarity of images and texts and reduce the contrastive loss:

$$L_{con} = -\sum_{x \in X} log \frac{exp(\phi(H_{c_x}, x))}{\sum_{i=1}^{m} (exp(\phi(H_{c_i}, x)))}, \qquad (3)$$

where $c_x$ is the true category of the image $x$, $m$ is the number of categories, and $\phi()$ represents the similarity score of text embeddings and image embeddings. The similarity score of the correct category should be higher than that of other categories.

## 4 Experiments

In this section, we first introduce the experimental settings. Then we evaluate our approach in the following four problem settings: 1) evaluating the quality of descriptions; 2) training-free zero-shot image classification; 3) few-shot prompt tuning image classification; 4) evaluating the explainable results.

### 4.1 Datasets and Settings

**Datasets** We employ 7 publicly available image classification datasets: Imagenet (Deng et al., 2009) OxfordPets (Parkhi et al., 2012), Flowers102 (Nilsback & Zisserman, 2008), Food101 (Bossard et al., 2014), FGVCAircraft (Maji et al., 2013), DTD (Cimpoi et al., 2014), and EuroSAT (Helber et al., 2019). These datasets constitute a comprehensive benchmark, which covers a diverse set of image distributions. For example, Food101, Flower102, and OxfordPets have images that are very common in daily life and dominant in CLIP training distributions. In addition, FGVCAircraft, DTD, and EuroSAT are less likely to appear in CLIP distributions and result in poor zero-shot performance. Imagenet, however, covers generic objects and fine-grained categories, which has a comprehensive neural distribution. Since we have distinctive descriptions for each class, the base-to-new setting cannot be applied for our models.

**Baselines** We compare our method with six existing baselines, including two zero-shot baselines and four prompt-tuning baselines. The zero-shot baselines are zero-shot CLIP (Radford et al., 2021) and CBD (Menon & Vondrick, 2023). Zero-shot CLIP (Radford et al., 2021) is the original CLIP model. For a fair comparison, we adopt the standard handcraft prompt as "an image of a [CLASS]". Since our method can be performed in the training-free setting, we compare our training-free version with these two training-free baselines. Additionally, the prompt tuning baselines are linear probe CLIP, COOP (Zhou et al., 2022b) COCOOP (Zhou et al., 2022a) and CLIP-Adapter Gao et al. (2024).

**samoyed**:
- snow-white pup,
- gentle giant,
- playful pooch,
- furry family member,
- cuddly cuddle buddy,
- smiling sammie,
- loyal companion,
- majestic majesty,
- happy-go-lucky pup

**rooster**:
- feathered friend,
- early riser,
- crowing companion,
- proud rooster,
- watchful guardian,
- feathered protector,
- barnyard beauty,
- cock-of-the-walk,
- barnyard buddy

**707-320**:
- passenger jet,
- long-range airliner,
- four-engine jet,
- iconic airliner,
- classic airliner,
- pioneering jet,
- pioneering aircraft,
- commercial airliner
- reliable workhorse

**cracked**:
- shattered lines,
- jagged edges,
- cracked lines,
- cracked surface,
- broken pieces,
- fractured fragments,
- splintered shards,
- shattered glass,
- broken shards

Figure 4: Examples of generated descriptions. We manually divided them into three types, namely features, alias, related concepts, which are shown in yellow, green, and black.

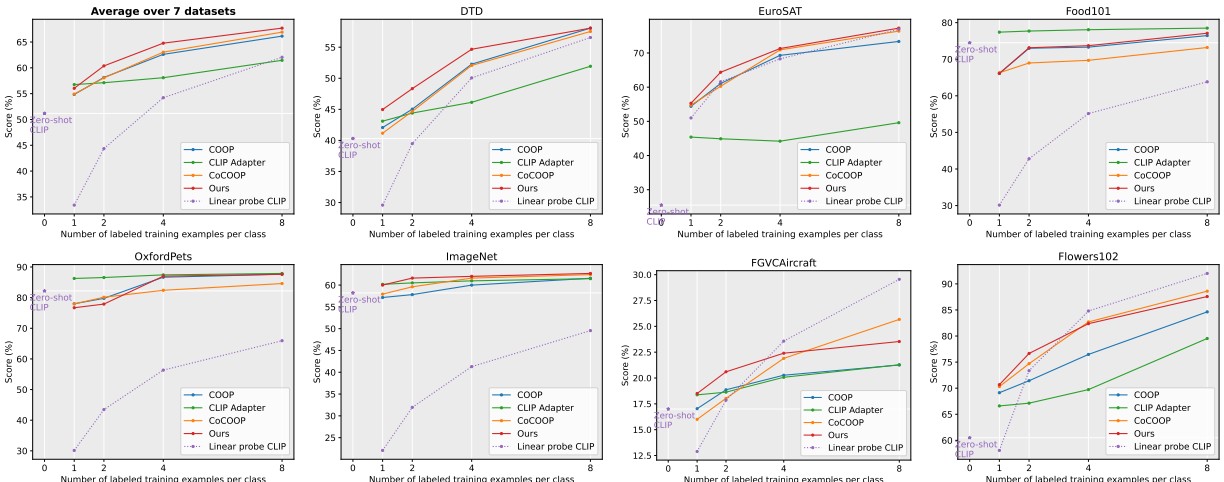

Figure 5: Main results of few-shot setting of our model on 7 datasets. We use RN50 as the CLIP backbone. Overall, our method can outperform other baselines. Comparing with the closest rival coop, we consistently outperform it over all datasets.

## 4.2 Implementation Details

For the description generation, we use GPT-4 and GPT-4o (OpenAI, 2023) to generate descriptions. For the prompt tuning and weighted aggregation method, we adopt 1/2/4/8-shot learning. We set $m, n$ as 4, so we have 8 shared learnable soft prompts. We initialize our learnable prompts by drawing from a zero-mean Gaussian distribution with standard deviation equal to 0.02. SGD is adopted as the optimizer, and an initial learning rate is set as 0.002, which is decayed by the cosine annealing rule. we use the warm-up trick by fixing the learning rate to 1e-5, as suggested in (Zhou et al., 2022b), only for the first epoch. We train 200 epochs for other datasets and 50 for Imagenet. To validate the generalization ability, we test our model on three CLIP backbones, RN50, VITB32, and VITB16. Our model is trained on an Nvidia A5000 GPU.

## 4.3 Description Generation

Figure 4 illustrates a selection of generated descriptions that span both frequent domains, such as animals, and less common domains, such as describable textures and fine-grained aircrafts. In studying these descriptions, we classified them into three main types: features, aliases, and related concepts.

The feature type encapsulates both visual and non-visual attributes pertinent to a category. They provide supplementary information that can facilitate a more comprehensive understanding of categories for the CLIP model. For instance, descriptions such as "shattered lines", "jagged edges", "cracked lines", and "cracked surface" pertain to the "crack" texture, capturing nuances related to the appearance of edges, the pattern

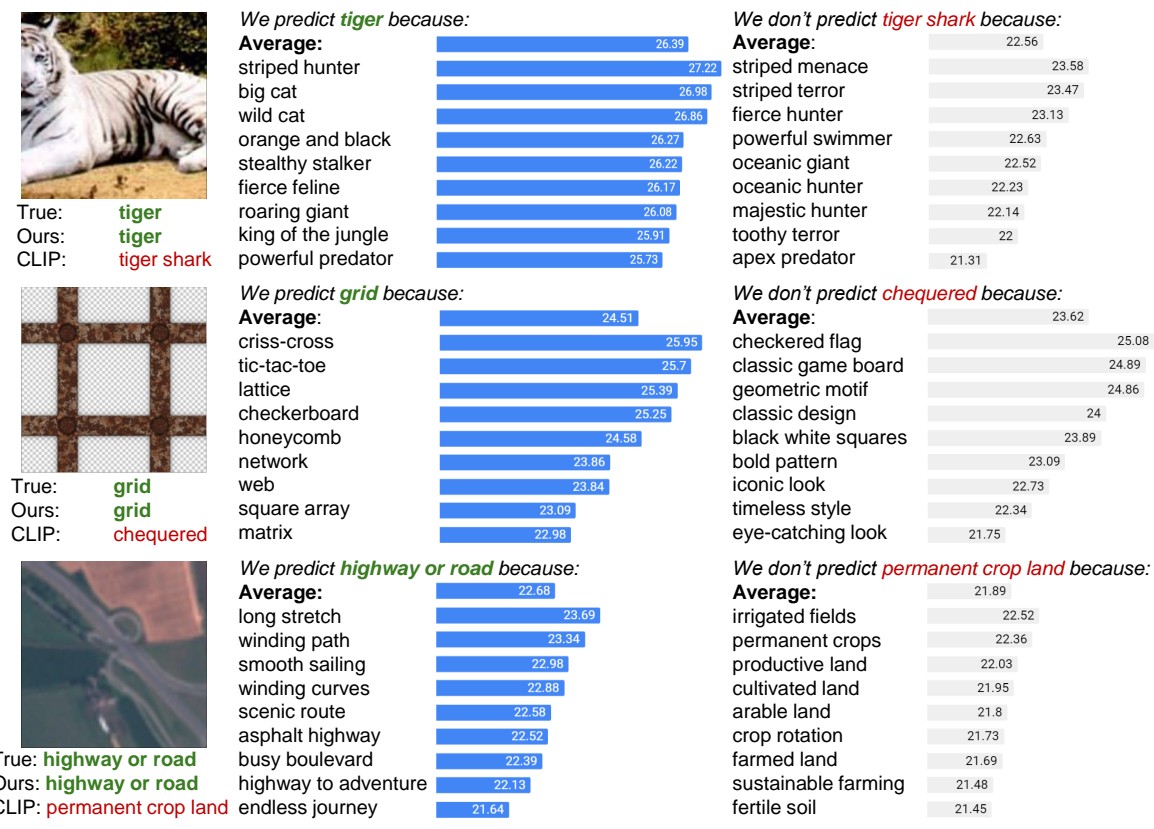

Figure 6: Examples of predictions from our training-free, zero-shot model (left, blue) and the CLIP model (right, grey). We provide visual comparisons of predictions from both CLIP and our models, with the ground truth labels presented beneath the corresponding images. The bar charts on the right illustrate the descriptions corresponding to each category, along with the similarity scores for these descriptions. Our predictions demonstrate accuracy, and the similarity scores for the descriptions provide evidence to support this claim, detailing why our model avoids selecting incorrect labels. These results also show that descriptions are beneficial to close the gap between images and texts.

of cracks, and the overall surface manifestation. This detailed feature helps the CLIP model understand the composition and semantics of a category. The alias type incorporates alternate terminology that is likely to appear in the image-text corpus. This broadens the label into multiple anchors, fostering a more comprehensive understanding. To illustrate, "707-320" refers to a specific aircraft model; however, without context, it could be misunderstood as a sequence of numbers or even a telephone number. By introducing aliases such as "iconic airliner" or "pioneering jet", we provide additional anchors, thus reducing potential misinterpretations by the CLIP model. Lastly, the related concept type includes common concepts that frequently occur in the image-text corpus. For example, a "rooster" is often associated with a "barnyard". This association allows the model to leverage the additional context provided by these concepts. In sum, our generated descriptions provide higher level interpretability and have closer distribution to image-text corpus, facilitating the CLIP model in better understanding and classifying images.

## 4.4 Training-free Zero-shot Image Classification

In the training-free setting, the weights of each description are the same, so each description will equally contribute to the classification. For each category, it has 9 to 10 descriptions. For each description, we construct the zero-shot prompt template as "an image of [CLASS], which relates to [DESCRIPTION]." We compare our model with two baseline models, namely CLIP (Radford et al., 2021) and CBD (Menon & Vondrick, 2023), across three different architectures: RN50, VITB32, and VITB16. From the Table 1, we can observe

that our training-free model consistently surpasses the performance of both baseline models across nearly all configurations and datasets. Specifically, our method showcases a consistent 3-5% improvement over the CLIP model, and also achieves 1-3% enhancement over CBD (Menon & Vondrick, 2023). Specifically, for certain datasets such as DTD (Cimpoi et al., 2014) and flower (Nilsback & Zisserman, 2008), our method has an improvement of around 6%. This consistent improvement suggests that augmenting prior knowledge and bringing the image-text distribution closer can promote image-text alignment in the CLIP embedding space, leading to superior image classification accuracy. Additionally, our method exhibits notable improvements in the fine-grained and less common domains such as flowers and descriptive textures. In contrast, more prevalent domains like pets show relatively smaller enhancements. This suggests that our approach effectively applies across diverse image domains. For categories characterized by larger domain gaps, our method assists the CLIP model in comprehending semantic information by attenuating the domain shift. Conversely, for more popular categories, the benefits of our approach may be marginal, as these categories inherently align closely with the image distribution. These observations substantiate our hypothesis that the prediction accuracy of VLMs is significantly influenced by the domain gap.

## 4.5 Few-shot Fine-grained Image Classification

In the few-shot domain prompt tuning setting, our objective is to further mitigate the distribution gap between images and descriptions, as there may exist some misalignment between them. To evaluate the effectiveness of our approach, we conducted experiments on seven different types of dataset, each representing a distinct image distribution. The results, as shown in Figure 5, consistently confirm the superior performance of our model over the most challenging rival CoCOOP across all settings. Specifically, we have around 2% average improvement to CoCOOP. However, CoCOOP outperforms our method on FGVCAircraft and Flower102 dataset with 16-shot experiments. It is expected since these labels are very similar providing little information, and the image specific prompt learning method could alleviate the problem. However, with the weighted aggregation strategy, the image distributions can better align with each label distribution and achieve better performance with respect to the average performance. This outcome reinforces that our techniques of description generation and reweighting mechanism facilitate alignment between text and image distributions within the CLIP latent embedding space. Remarkably, our method exhibits even greater improvements on fine-grained and uncommon image datasets, such as DTD, and EuroSAT, compared to popular image datasets. This observation highlights the robustness of our approach when dealing with significant distribution discrepancies. This implies that our model effectively leverages prior knowledge and aligns distributions more accurately, enabling enhanced performance on datasets with larger distribution gaps.

## 4.6 Explainable Qualitative Results

Since we have incorporated additional textual descriptions, we can interpret predictions based on the similarity between each description and the image. Figure 6 provides illustrative examples in different image types. We showcase some images from various domains including texture images (Cimpoi et al., 2014), satellite images (Helber et al., 2019), and animal images (Deng et al., 2009). These examples demonstrate how additional information influences the final decision.

Consider an instance from the prevalent animal domain, where we encounter an image of a tiger. In the original CLIP model, the prediction might be incorrect due to the similarities between a tiger and a tiger shark at the textual level. However, by expanding the label with descriptions, CLIP gains a better understanding of the actual label. As we can see in the examples, descriptions of "striped hunter," "big cat," and "wild cat" exhibit relatively higher similarity to the image, correctly indicating the animal's classification as a tiger. In contrast, descriptions associated with the tiger shark, such as "oceanic," result in a lower similarity to the image, correctly distinguishing it as a different category.

On the other hand, as for the uncommon domains, our descriptions also provide some precise information. For example, the "criss-cross" show the features of the grid texture and meanwhile using the image-text manner to describe it. As a result, it obtains the highest similarity score, and thus helps the model to make the right prediction.

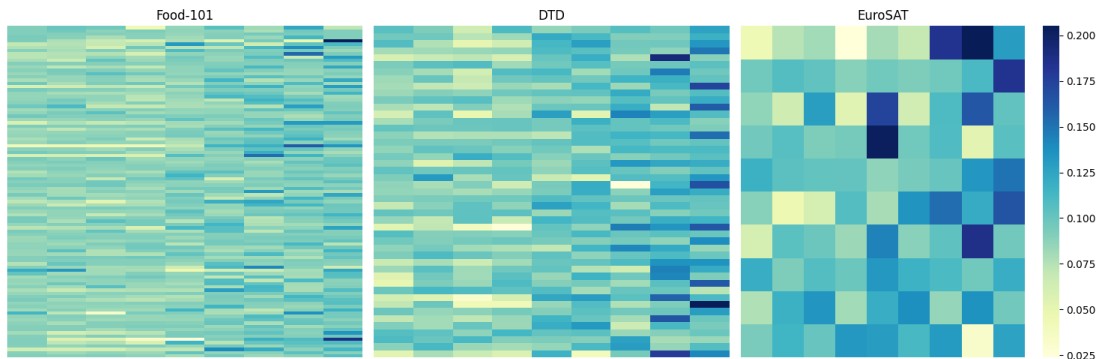

Figure 7: Example weights for the Food-101, DTD and EuroSAT datasets in the 1-shot setting. Each cell corresponds to one description. Specifically, each row represents a category, and typically, each category has nine descriptions. The darker the cell, the greater the weight, reflecting the importance of the description in distinguishing the category.

These examples demonstrate how our approach enables CLIP to consider multiple pieces of evidence to make more accurate predictions. By leveraging descriptions and expanding the label space, CLIP gains a deeper insight into the visual content and context, allowing for improved classification performance across various types and domains of images.

### 4.7 Explainable Weighted Aggregation

The heatmap shown in Figure 7 illustrates the varying levels of importance among different descriptions, highlighting the effectiveness of our weighted aggregation module. The most critical descriptions are predominantly found on the right side of the heatmap, which aligns with the caption descriptions. This pattern indicates a strong link between the textual captions and the visual distribution of images, which plays a significant role in determining classification confidence. However, the general descriptions associated with uncommon datasets and categories carry smaller weights, indicating that these descriptions are less significant and may overlap with those of other categories. In addition, the weights assigned to the descriptions provide insight into the distinctive features and distributions unique to each category in the dataset. For example, in the EuroSAT dataset, descriptions like "rectangular shapes" and "pinkish color tones" are keys to identifying annual crop land, whereas a "dark background" is a significant indicator of the forest category. This aspect underscores its explainability, offering users a clear understanding of how it discerns and categorizes images. It also proves that the description is capable of aligning the image distribution.

### 4.8 Robustness Evaluation

We evaluate the robustness of our model using three different CLIP backbones: RN50, VITB32, and VITB16. To assess its generalization ability on uncommon domains, we test it on three fine-grained datasets: DTD (Cimpoi et al., 2014), EuroSAT (Helber et al., 2019), and FGVCAircraft (Maji et al., 2013). The results, presented in Table 2, highlight the consistent improvements achieved by our model. Across all datasets and backbone architectures, our method outperforms both the baseline CLIP and COOP models. Specifically, on DTD, our model achieves a performance gain of up to 7.3% over COOP when using RN50. A similar trend is observed for VITB32 and VITB16, where our method improves over COOP by 4.4% and 2.5% on average, respectively. Similarly, for EuroSAT, our model significantly outperforms baselines. In the challenging FGVCAircraft dataset, where fine-grained differences among classes make classification particularly difficult, our model again consistently outperforms baselines. For example, with RN50 and 8-shot training, our approach improves over COOP by 10.5%. Notably, on VITB16, our method shows the largest gain, reaching an improvement of 19.3%. Compared to the baseline CLIP model, our model demonstrates a considerable improvement in performance, validating our model's domain adaptation capability. The relatively poor performance of the CLIP model illustrates the substantial distribution discrepancy between

| model | shots | DTD | | | EuroSAT | | | FVGCAircraft | | |
|---|---|---|---|---|---|---|---|---|---|---|
| | | CLIP | COOP | Ours | CLIP | COOP | Ours | CLIP | COOP | Ours |
| RN50 | 1 | 40.30 | 42.07 | **44.97** | 25.50 | 54.40 | **55.30** | 17.00 | 17.03 | **18.50** |
| | 2 | 40.30 | 45.03 | **48.33** | 25.50 | 61.07 | **64.37** | 17.00 | 18.87 | **20.60** |
| | 4 | 40.30 | 52.27 | **54.67** | 25.50 | 68.77 | **71.43** | 17.00 | 20.26 | **22.40** |
| | 8 | 40.30 | 58.07 | **59.73** | 25.50 | 73.37 | **77.27** | 17.00 | 21.30 | **23.53** |
| VITB32 | 1 | 43.80 | 44.87 | **47.80** | 38.40 | 51.33 | **54.63** | 18.60 | 20.26 | **20.80** |
| | 2 | 43.80 | 48.07 | **50.07** | 38.40 | 60.57 | **62.47** | 18.60 | 21.23 | **22.53** |
| | 4 | 43.80 | 54.17 | **55.57** | 38.40 | 69.90 | **70.67** | 18.60 | 22.80 | **26.37** |
| | 8 | 43.80 | 59.87 | **61.20** | 38.40 | 74.40 | **75.50** | 18.60 | 23.90 | **28.26** |
| VITB16 | 1 | 44.40 | 47.26 | **49.53** | 40.90 | 55.03 | **56.57** | 23.00 | **27.46** | 25.50 |
| | 2 | 44.40 | 53.13 | **54.83** | 40.90 | 66.13 | **66.36** | 23.00 | 28.67 | **29.73** |
| | 4 | 44.40 | 59.03 | **59.30** | 40.90 | 74.07 | **74.80** | 23.00 | 29.43 | **33.40** |
| | 8 | 44.40 | 63.90 | **65.03** | 40.90 | **77.97** | 77.00 | 23.00 | 32.20 | **38.43** |

Table 2: Comparison of our prompt tuning model on few-shots settings with COOP (Zhou et al., 2022b). We test our model on three major uncommon datasets with three CLIP backbones to validate our robustness and effectiveness. Compared to baselines, our results show consistent improvement across all settings. The number indicates the image classification accuracy. Higher is better. The best results are shown in **bold**.

| Dataset | method | 1shot | 2shot | 4shot | 8shots |
|---|---|---|---|---|---|
| DTD | CLIP | 40.30 | 40.30 | 40.30 | 40.30 |
| | descrption | 42.00 | 42.00 | 42.00 | 42.00 |
| | descrption+prompt | 44.33 | 48.07 | 54.00 | 57.83 |
| | descrption+prompt+weight | **44.97** | **48.33** | **54.67** | **58.07** |
| EuroSAT | CLIP | 25.50 | 25.50 | 25.50 | 25.50 |
| | descrption | 26.50 | 26.50 | 26.50 | 26.50 |
| | descrption+prompt | 54.96 | 62.46 | 70.93 | 77.06 |
| | descrption+prompt+weight | **55.30** | **64.37** | **71.30** | **77.27** |
| FVGCAircraft | CLIP | 17.00 | 17.00 | 17.00 | 17.00 |
| | descrption | 19.60 | 19.60 | 19.60 | 19.60 |
| | descrption+prompt | 18.30 | 20.40 | 21.90 | 23.53 |
| | descrption+prompt+weight | **18.50** | **20.60** | **22.40** | **25.26** |

Table 3: Ablation study of each components.

images and texts. These consistent improvements across various domains demonstrate our model's ability to effectively bridge the distribution gap between images and textual descriptions through learnable prompts, even in cases where the discrepancy is huge. Furthermore, our model surpasses the performance of the related baseline model, showcase the efficacy of our weighted strategy. The consistent improvements across domains validate the robustness of our method in diverse and challenging environments.

## 4.9 Ablation Study

We conduct an ablation study to evaluate the impact of each component in our method. The results in Table 3 show that each component progressively enhances performance.

From the experiments, we observe that each component has a positive impact on the classification. For example, for the DTD dataset, the base method (CLIP) achieved a consistent score of 40.30 across all shots. Adding a description component improves the score slightly to 42.00. The addition of a prompt to the description further increases the performance significantly, reaching 44.33, 48.07, 54.00, and 57.83 for 1-shot, 2-shot, 4-shot, and 8-shot learning, respectively. By adding a weight component to the description+prompt,

we observe the best performance of 44.97, 48.33, 54.67, and 58.07 for 1-shot, 2-shot, 4-shot, and 8-shot learning, respectively. A similar trend is observed in EuroSAT and FGVCAircraft, the introduction of prompts leads to significant gains and the weighted approach further refines performance. In summary, the addition of a "description", "prompt", and "weight" components consistently improved the performance across all datasets and learning scenarios. This demonstrates the effectiveness of each component in our proposed method.

### 4.10 Effect of the number of descriptions

In this section, we analyze the impact of the number of descriptions on the performance of the proposed method across three different datasets: DTD, EuroSAT, and FVGCAircraft. The results are summarized in the Table 4.

For the DTD dataset, the performance with three descriptions was 40.20, 47.07, 53.10, and 57.33 for 1-shot, 2-shot, 4-shot, and 8-shot learning, respectively. Increasing the number of descriptions to six led to a slight improvement in performance, with scores of 42.70, 48.23, 53.97, and 57.77 for the respective shot learning scenarios. The best performance was achieved with nine descriptions, which resulted in scores of 44.97, 48.33, 54.67, and 58.07. Similar to the EuroSAT and FVGCAircraft dataset, the performance improves slightly as descriptions increases. In summary, increasing the number of descriptions consistently improved the performance across all datasets and learning scenarios. This indicates that using more descriptions can effectively enhance the accuracy of the proposed method in various few-shot learning scenarios.

| Dataset | # descriptions | 1-shot | 2-shot | 4-shot | 8-shot |
|---------|---------------|--------|--------|--------|--------|
| DTD | 3 | 40.20 | 47.07 | 53.10 | 57.33 |
| | 6 | 42.70 | 48.23 | 53.97 | 57.77 |
| | 9 | **44.97** | **48.33** | **54.67** | **58.07** |
| EuroSAT | 3 | 54.30 | 61.57 | 70.77 | 75.20 |
| | 6 | 54.70 | 62.40 | 71.20 | 77.23 |
| | 9 | **55.30** | **64.37** | **71.30** | **77.27** |
| FVGCAircraft | 3 | 16.67 | 18.70 | 21.26 | 22.10 |
| | 6 | 17.10 | 19.80 | 21.90 | 25.00 |
| | 9 | **18.50** | **20.60** | **22.40** | **25.26** |

Table 4: Results of effect of the number of descriptions.

## 5 Limitation

While Language Learning Models (LLMs) exhibit impressive capability in generating semantic information, there are nonetheless certain limitations that influence their outcome, particularly in the context of fine-grained image classification. For example, when encountering classes that demand intricate descriptions, LLMs occasionally fall short in generating meaningful descriptions. Instead, they tend to rendering generic descriptions that offer minimal distinctive information for fine-grained categorization. For instance, in OxfordPets dataset (Parkhi et al., 2012), around 20% categories have the description saying "man's best friend" which provide almost none additional information. One possible solution is providing additional contextual information specific to the domain under consideration. By doing so, LLMs could potentially filter out redundant descriptions, thereby enhancing their ability to generate more detailed and distinguishing descriptions. The exploration of methods to improve LLMs' performance in these more complex scenarios represents an interesting direction for future research.

## 6 Conclusion

In this paper, we propose an improved prompting method for vision-language models, enhancing both the accuracy and interpretability of few-shot fine-grained image classification. Specifically, we introduce a novel framework for zero-shot classification that aligns vision-language models with domain-specific distributions through soft prompt tuning and weighted aggregation. Extensive experiments on multiple benchmark datasets demonstrate that our approach consistently outperforms baseline methods in classification accuracy while maintaining interpretability. By leveraging interpretable descriptions, our model effectively bridges the semantic gap between textual prompts and visual features, leading to improved domain alignment. The integration of weighted aggregation further refines this alignment, optimizing performance without compromising interpretability.

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
