# OpenReview forum: "Few-shot Fine-grained Image Classification with Interpretable Prompt Learning through Distribution Alignment"
_TMLR — Withdrawn by Authors_

### Review · Reviewer_hHFV · 2025-03-05

**Summary Of Contributions:**

The paper claims to develop a method for fine-grained image classification. The authors effectively delineate the disparity that exists between the assigned labels and the actual visual content of the images. To bridge this gap, they propose generating detailed text descriptions and employing prompt tuning techniques.

**Audience:**

Yes

**Broader Impact Concerns:**

There are no ethical concerns.

**Claims And Evidence:**

Yes

**Requested Changes:**

1. Increase the comparison of different methods.
2. Add more explanations about the effectiveness of the method.

**Strengths And Weaknesses:**

pros:
1. The work comes up with a simple but relatively involved method for CLIP's performance on image classification.
2. The paper is easy to follow.
3. Adequate experiments illustrate the effectiveness of the method.

cons:
1. In the few-shot experiment, it is mentioned that GPT-4o is used to generate textual information for images. If we compare this with directly using GPT-4o to complete the classification, will the effect be better?
2. The method for establishing text descriptions proposed in this paper is very similar to this one[1]. Could you further analyze the impacts of different types of text descriptions on the classification effect, such as those mentioned in the paper, namely "features", "alias", and "related concepts"?
3. Insufficient comparison with the recent related methods. To be a fair comparison, this reviewer believes they need to compare with the recent related methods[1],[2]

[1] Yi, Chao et al. “Leveraging Cross-Modal Neighbor Representation for Improved CLIP Classification.” 2024 IEEE/CVF Conference on Computer Vision and Pattern Recognition (CVPR) (2024): 27392-27401.

[2] Pratt, Sarah et al. “What does a platypus look like? Generating customized prompts for zero-shot image classification.” 2023 IEEE/CVF International Conference on Computer Vision (ICCV) (2022): 15645-15655.

---

### Review · Reviewer_juex · 2025-03-06

**Summary Of Contributions:**

This work proposes to use LLMs and MLLMs to enhance fine-grained image classification and their interpretability, by generating descriptions and textual features of images and class labels. The interpretability here is the contribution of each description to the image (similarity score). They also use soft prompt learning to enhance the performance. Finally, they learn a matrix to aggregate the text embedding. This matrix provides the weights for each embedding of the description. However, it remains unclear how this combination of techniques offers a distinct contribution, as all elements have been proposed and well-explored in previous literature.

**Audience:**

No

**Broader Impact Concerns:**

No ethical concerns.

**Claims And Evidence:**

No

**Requested Changes:**

Overall, I believe the paper does not currently align well with TMLR's standards and would benefit from substantial, core revisions. Consequently, I do not have specific adjustments to recommend, as the work appears to require a major, core change.

**Strengths And Weaknesses:**

Strengths: It was challenging to identify any clear strengths in the submission.

Weaknesses:
- The proposed work appears to be exactly like VCD [R2] combined with prompt learning from COOP. VCD [R2] is known for its training-free and interpretability approach, and COOP introduces prompt tuning. The current submission does not seem to offer any new contributions.
- The introduction lacks a clear, motivating narrative. The overall presentation does not provide a coherent storyline, and the origin of Figure 1 is unclear. It is not evident whether this figure is based on a hypothesis (e.g., imagination) or derived from empirical data. How are the authors "proving" that plain text distribution is different from the image-text corpus distribution?
- The paper does not introduce notable innovation. The utilization of LLMs to generate descriptions has become a standard practice in the domain of CLIP prompt learning and adaptation, as seen in works like CuPL [R1] and VCD [R2] and many others. Also, the performance improvement is very much expected and not surprising compared to baselines as the authors incorporate a Multimodal LLM while baseline do not. This is not a fair comparison, and moreover, the improvement comes on the expense of an expensive (image-dependant) Multimodal GPT. Unlike CuPL and similar approaches that only need to store fixed class descriptions as linear classifier weights—thus removing the need for LLMs during inference—this paper's method still relies on an additional core model, the Multimodal LLM, along with CLIP, because it now depends on image-specific information rather than static class definitions.
- The experimental evaluation is very limited, with comparisons made only against COOP—the initial work in CLIP prompt learning. Since Coop, there has been an abundance of works and recent advances in CLIP adaptation and prompt learning. In fact, after Coop this has been a new research direction, and several methods have been developed that outperform COOP, and even without requiring a Multimodal GPT, including training-free approaches like CuPL and VCD. The absence of comparisons with these more recent methods further diminishes the overall impact of the work.

[R1] Generating Customized Prompts for Zero-Shot Image Classification
[R2] Visual Classification via Description from Large Language Models

---

### Review · Reviewer_vEYM · 2025-03-10

**Summary Of Contributions:**

This paper proposes a hybrid prompt augmentation technique that leverages reasoning visual descriptions from VLMs and semantic knowledge from pure LLMs. The empirical results show that the proposed method can clearly improve over the baselines and previous methods, demonstrating the effectiveness of this method. Technical contribution of this work includes more hybrid prompt augmentation techniques and a weighting mechanism. Besides, this work also present many forms of visualization results to validate that method is interpretable to some extent.

**Audience:**

No

**Broader Impact Concerns:**

I consider that the current version of this work does not unveil significant novel discoveries and knowledge, and thus has limited contributions to the mature prompt augmentation field. I would suggest the author enhance this work for in-depth investigations for another round instead of purely using VLM to augment prompts based on existing techniques to improve the performance.

**Claims And Evidence:**

No

**Requested Changes:**

More experiments and investigations. Please refer to weaknesses.

**Strengths And Weaknesses:**

Strengths:

- The overall presentation quality of this paper is good.
- The proposed method incorporates the generated texts from the latest VLM, e.g., GPT-4o, which I think is touched by few studies.
- The performance improvements over the baselines are clear.
- The entire paper is clear and easy-to-follow.


Weaknesses:

Unsupported claim: In Fig. 1, how does the author assert the relative distributional distance of three types of text descriptions w.r.t. the image distribution. More empirical evidence is needed.

Furthremore, it lacks in-depth investigations and analysis:

- Needs evaluation of the effectiveness of the proposed approach on long-context CLIP models [1].
- Need to present some qualitative hallucination cases of generated text descriptions.
- Lack sensitivity analysis on the generated descriptions.
- Missing references and baselines: lack references to the most recent works which also leverages the LLM and foundation models for the prompt augmentation [2, ]. These methods, though not targeting the exactly the same problem, are highly related and easy-to-adapt and thus is highly recommended to be compared under this setting (even without fine-tuning).
- The in-depth category-level investigation of performance changes is needed when using the different types of prompt augmentation techniques.
- How would the quality of the generated texts affect performance in the model specific level? If include the results on the other VLM generated texts, the reader will be clearer about the line of the proposed VLM-informed prompt augmentation method.

About the improved presentation:

- For table 3, it would be better and clearer to include hte performance improvements w.r.t. the previous row when ablating the method.
- Needs qualitative results for main comparisons.
- The further investigation on the scaled number of descriptions can be insightful, e.g., when the performance will saturate?

About the improved writing:

- the overall writing of this paper is good, but some nuanced places can be further refined, e.g., ‘The similarity score of the correct category should be higher than that of other categories.’ would be more formal if changed into ‘The similarity score of the correct category is expected be higher than that of other categories.’

[1] **LLM2CLIP: Powerful Language Model Unlocks Richer Visual Representation**

[2] Large Language Models are Good Prompt Learners for Low-Shot Image Classification

[3] Waffling around for Performance - Visual Classification with Random Words and Broad Concepts

[4] PERCEPTIONCLIP: VISUAL CLASSIFICATION BY INFERRING AND CONDITIONING ON CONTEXTS

[5] Improving CLIP Training with Language Rewrites

[6] Learning Hierarchical Prompt with Structured Linguistic Knowledge for Vision-Language Models

---

### Note · Authors · 2025-03-22

I have read and agree with the venue's withdrawal policy on behalf of myself and my co-authors.